# Pay-it-forward influenza vaccination among older adults and children: A cost-effectiveness analysis in China

**Fanny Fong-Yi Tang**[1]*, **Priya Kosana**[2], **Mark Jit**[3], **Fern Terris-Prestholt**[4,5], **Dan Wu**[6,7], **Jason J. Ong**[6,8], **Joseph D. Tucker**[6,9]

1 Li Ka Shing Faculty of Medicine, University of Hong Kong, Hong Kong, SAR, China, 2 School of Public Health, Yale University, New Haven, CT, United States of America, 3 Department of Infectious Diseases Epidemiology, London School of Hygiene and Tropical Medicine, London, United Kingdom, 4 UNAIDS, Geneva, Switzerland, 5 Department of Global Health and Development, Faculty of Public Health and Policy, London School of Hygiene and Tropical Medicine, London, United Kingdom, 6 Department of Clinical Research, Faculty of Infectious and Tropical Diseases, London School of Hygiene and Tropical Medicine, London, United Kingdom, 7 Nanjing Medical University, Nanjing, China, 8 Melbourne Sexual Health Centre, Monash University, Melbourne, Australia, 9 Institute for Global Health and Infectious Diseases, Department of Medicine, University of North Carolina at Chapel Hill, Chapel Hill, NC, United States of America

* fanitang@connect.hku.hk

**Data Availability Statement:** Researchers with proposed use of the data can make a data request to the corresponding author with specific data needs, analysis plans and dissemination plans.

## Abstract

A quasi-experimental study was conducted to evaluate the effectiveness of a pay-it-forward strategy for increasing influenza vaccination among children and older adults compared to a self-paid vaccination strategy in China. Pay-it-forward is an innovative community-engaged intervention in which participants receive a free influenza vaccination and are then asked if they would like to donate or create a message to support subsequent vaccinations. This economic evaluation used a decision-tree model to compare pay-it-forward to a standard of care arm in which patients had to pay for their own influenza vaccine. The analysis was performed from the healthcare provider perspective and costs were calculated with 2020 United States dollars. The time horizon was one year. In the base case analysis, pay-it-forward was more effective (111 vs 55 people vaccinated) but more costly than standard-of-care ($4477 vs $2725). Pay-it-forward spurred 96.4% (107/111) of individuals to voluntarily donate to support influenza vaccination for high-risk groups in China. Further costing and implementation research is needed to inform scale up.

## Introduction

Influenza causes considerable morbidity and mortality worldwide, with the highest burden among adults aged over 60 years old and children aged from 6 months to 8 years [1–5]. Influenza vaccination is the most effective way to avert influenza-related mortality and morbidity [6, 7], but the vaccination rate in China remains low. More than three-quarters of older adults and children have never received an influenza vaccination [8]. Coverage is similar or even lower in many low and middle-income countries [8–10]. These data suggest the need for implementation strategies to increase influenza vaccination uptake.

**Funding:** This research is supported by the National Institutes of Health (JT: R01AI158826) and The Bill & Melinda Gates Foundation (JT: OPP1217240). The funders had no role in study design, data collection and analysis, decision to publish, or preparation of the manuscript. The publication of study results was not contingent on the sponsor's approval or censorship of the manuscript.

**Competing interests:** The authors have declared that no competing interests exist.

Pay-it-forward is a social innovation in which a person receives a gift, then voluntarily gives a gift to another person (Fig 1). The benefits of pay-it-forward programmes include reducing barriers due to cost and mobilising community altruism, which may be useful for increasing uptake of community health services [11, 12]. Experimental studies demonstrate that prosocial behaviours can cascade through social networks [13, 14], allowing pay-it-forward to leverage individual human connections to bring about wider community health benefits.

Few studies have examined pay-it-forward interventions [9], underscoring the importance of investigating its implementation. There have also been few cost-effectiveness studies on social innovations broadly, and on pay-it-forward in particular [15]. There are also no studies examining the financial sustainability of pay-it-forward.

We conducted a three-arm quasi-experimental study to assess the effectiveness of a pay-it-forward strategy to increase influenza vaccine uptake in China compared to standard-of-care and free vaccination strategies [16]. Here we describe a cost-effectiveness and financial sustainability analysis of the intervention conducted alongside the original study.

## Methods

### Parent effectiveness study

In the parent study, three research sites in Guangdong Province, China were selected based on their urbanicity and average income level. The three study sites were rural (Yangshan County in Qingyuan city), suburban (Zengcheng District), and urban (Tianhe district in Guangzhou City). Each site had clinics which sold influenza vaccines for a fee. The selected primary care clinics had sufficient influenza vaccines stock and properly trained medical staff (nurses, doctors) familiar with influenza vaccination. In each study site, 50 eligible participants were

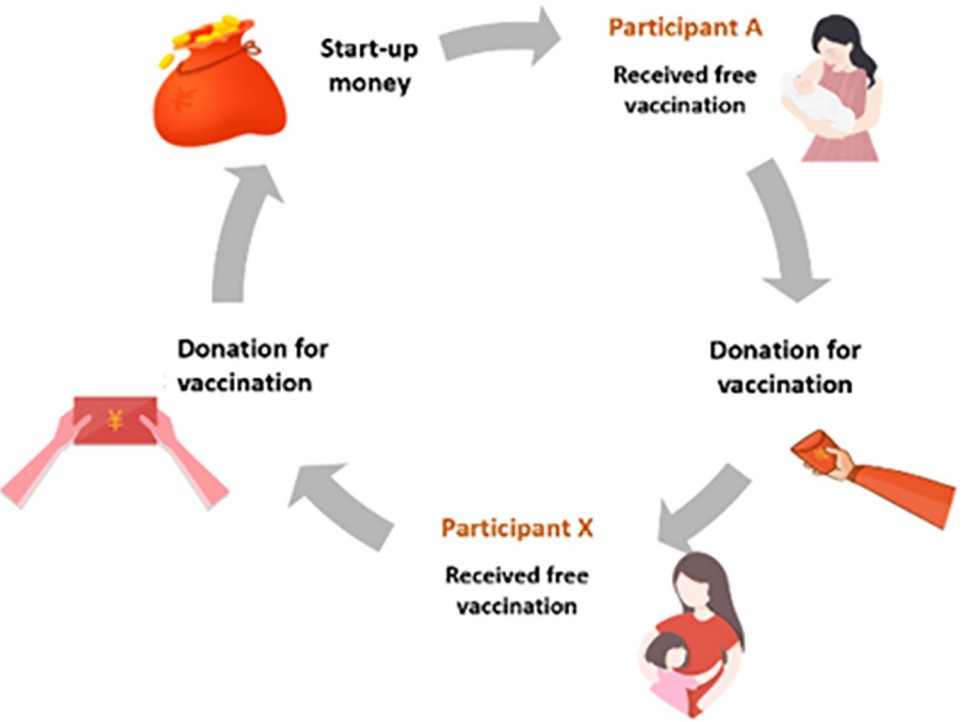

**Fig 1. Flowchart showing overview of pay-it-forward.**

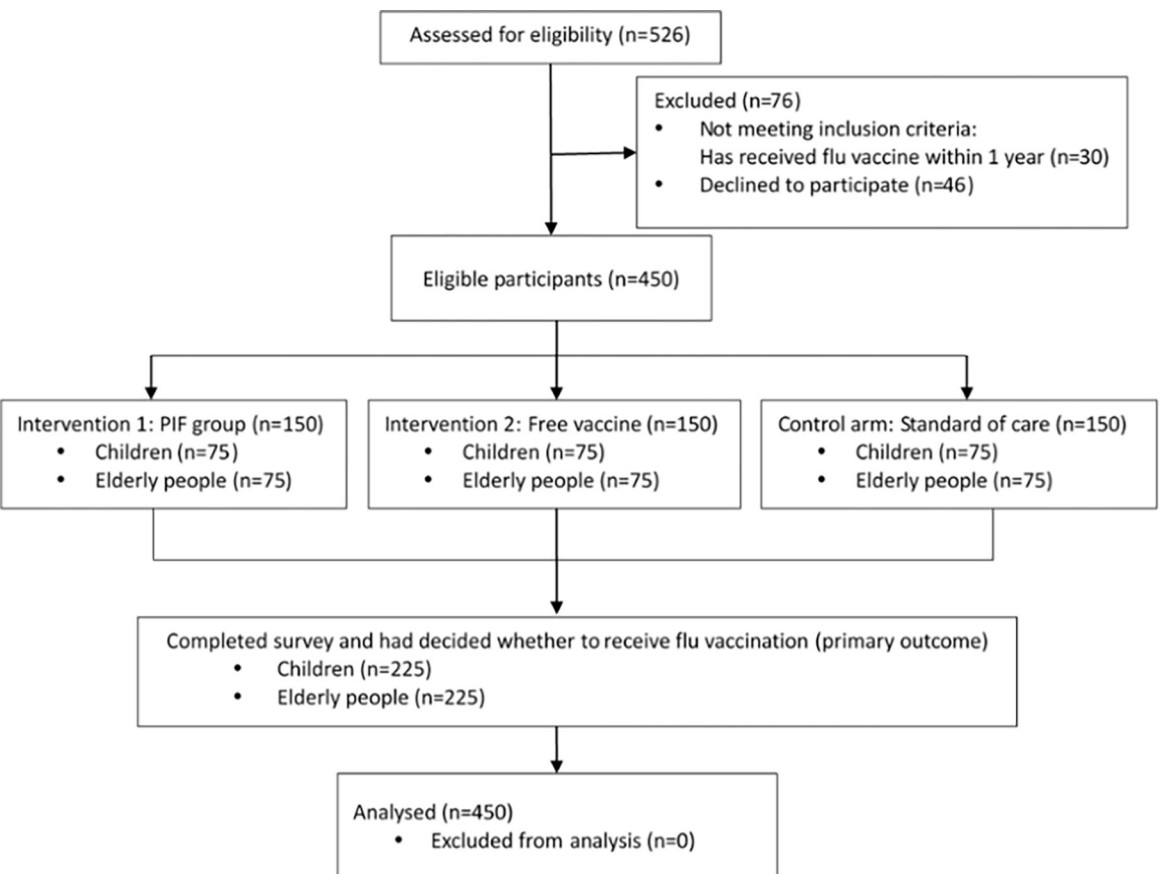

**Fig 2. CONSORT flowchart of parent study.**

recruited into each of the three study arms, i.e. the pay-it-forward arm, the standard-of-care arm, and the free vaccination arm (Fig 2).

150 participants were recruited at each study site and a total of 450 participants were recruited into the study. Eligible individuals were between six months and eight years old (children) or ≥ 60 years old (older adults). The participants had no acute moderate or severe illnesses and were eligible to receive an influenza vaccine based on clinical evaluation from a physician. A questionnaire was given to all the participants after recruitment in order to record demographic data about the study population (S1 Appendix).

In the standard-of-care arm, staff provided participants an introductory pamphlet about influenza vaccination, and asked if they were willing to pay out-of-pocket to receive influenza vaccination. The brand, and hence the cost of the influenza vaccine offered by the clinic was different for each study site and is listed in Table 1. In the pay-it-forward arm, participants were given the same pamphlet about influenza vaccination and a study coordinator explained the pay-it-forward program using handwritten postcards developed by other participants. They were not charged for the vaccine. Among those who received the vaccine, they were asked about donating money or creating a postcard message for a subsequent person. Donations were completely voluntary and fixed amounts for donation were suggested to the participant at 50, 100, 150 and 200 Renminbi (RMB) (US$7.49, $15.0, $22.5, $29.9, respectively), with an additional 'other' option for participants who wanted to donate another amount. Participants in the free vaccination arm were provided with the same introductory information

**Table 1. Unit costs (in 2020 USD\*) and frequency of vaccine use.**

| Intervention | Cost item | Unit cost | Resource use | Source |
|---|---|---|---|---|
| ***Pay-it-forward*** | | Staff wage per hour (USD/hour)‡ | | |
| • Start-up costs | | | | |
| | *Time cost of designing postcards to be written on by participants in the PIF programme†1* | 7.47 | 1 x 5 hr | Personal communication with research staff |
| | *Time cost of research fellow in preparatory workshop†* | 12.7 | 1 x 5 hr | Personal communication with research staff |
| | *Time cost of research assistant in preparatory workshop†* | 7.47 | 1 x 5 hr | Personal communication with research staff |
| | *Time cost of nurses in preparatory workshop* | 4.81 | 3 x 1 hr | Personal communication with research staff China Social Welfare Foundation [17] |
| | *Time cost of clinic coordinators in preparatory workshop* | 4.81 | 3 x 1 hr | Personal communication with research staff China Social Welfare Foundation [17] |
| • Fixed costs | | | | |
| | *Cost of hiring vaccinators (doctors) for the three clinics* | 11.07 | 3 x 70 hr | China Social Welfare Foundation [17] |
| • Recurrent costs | | | | |
| | *Time cost of nurses in recruiting patients to join the PIF programme††* | 4.81 | 3 x 25 hr | Personal communication with research staff China Social Welfare Foundation [17] |
| | *Time cost of clinic coordinators in performing administrative work for the PIF programme††* | 4.81 | 3 x 25 hr | Personal communication with research staff China Social Welfare Foundation [17] |
| | | Cost per vaccine (USD) | | |
| | *Cost of adult vaccines in Yangshan#* | 22.9 | 41 | Health clinic reimbursement invoices |
| | *Cost of adult vaccine in Zengcheng* | 22.9 | 17 | Health clinic reimbursement invoices |
| | *Cost of child vaccine in Zengcheng* | 8.38 | 25 | Health clinic reimbursement invoices |
| | *Cost of adult vaccine in Tianhe#* | 12.2 | 28 | Health clinic reimbursement invoices |
| | | Cost per item / batch (USD) | | |
| | *Introductory pamphlets for the PIF programme (batch cost)* | 29.0 | 1 | Project research staff invoices |
| | *Ballpoint pens (for writing messages on PIF postcards) (batch cost)* | 2.25 | 1 | Project research staff invoices |
| | *Cost of printing postcards (batch cost)* | 7.41 | 1 | Project research staff invoices |
| | *Surgical gloves* | 0.075 | 222 | Personal communication with research staff |
| **Intervention** | **Cost item** | **Unit cost** | **Resource use** | **Source** |
| ***Standard-of-care*** | | Staff wage per hour (USD/hour)\* | | |
| • Start-up costs | *Time cost of nurses in preparatory workshop* | 4.81 | 3 x 1 hr | China Social Welfare Foundation [17] |
| | *Time cost of clinic coordinators in preparatory workshop* | 4.81 | 3 x 1 hr | China Social Welfare Foundation [17] |
| • Fixed costs | | | | |
| | *Cost of hiring vaccinators for the three clinics* | 11.07 | 3 x 70 hr | China Social Welfare Foundation [17] |
| • Recurrent costs | | | | |
| | *Time cost of nurses in recruiting patients to receive seasonal influenza vaccination††* | 4.81 | 3 x 12.5 hr | Personal communication with research staff China Social Welfare Foundation [17] |
| | *Time cost of clinic coordinators in performing administrative work for influenza vaccination††* | 4.81 | 3 x 12.5 hr | Personal communication with research staff China Social Welfare Foundation [17] |
| | | Cost per vaccine (USD) | | |

(*Continued*)

**Table 1.** (Continued)

| | | | | |
|---|---|---|---|---|
| | *Cost of adult vaccines in Yangshan#* | 22.9 | 15 | Health clinic reimbursement invoices |
| | *Cost of adult vaccine in Zengcheng* | 22.9 | 5 | Health clinic reimbursement invoices |
| | *Cost of child vaccine in Zengcheng* | 8.38 | 14 | Health clinic reimbursement invoices |
| | *Cost of adult vaccine in Tianhe#* | 12.2 | 21 | Health clinic reimbursement invoices |
| | | Cost per item (USD) | | |
| | *Surgical gloves* | 0.075 | 110 | Personal communication with research staff |
| **Intervention** | **Cost item** | **Unit cost** | **Resource use** | **Source** |
| ***Free vaccination*** | | | | |
| • Start-up costs | | Staff wage per hour (USD/hour)* | | |
| | *Time cost of nurses in preparatory workshop* | 4.81 | 3 x 1 hr | Personal communication with research staff China Social Welfare Foundation [17] |
| | *Time cost of clinic coordinators in preparatory workshop* | 4.81 | 3 x 1 hr | Personal communication with research staff China Social Welfare Foundation [17] |
| • Fixed costs | | | | |
| | *Cost of hiring vaccinators for the three clinics* | 11.07 | 3 x 70 hr | China Social Welfare Foundation [17] |
| • Recurrent costs | | | | |
| | *Time cost of nurses in recruiting patients to join the PIF programme††* | 4.81 | 3 x 12.5 hr | Personal communication with research staff China Social Welfare Foundation [17] |
| | *Time cost of clinic coordinators in performing administrative work for the PIF programme††* | 4.81 | 3 x 12.5 hr | Personal communication with research staff China Social Welfare Foundation [17] |
| | | Cost per vaccine (USD) | | |
| | *Cost of adult vaccines in Yangshan#* | 22.9 | 42 | Health clinic reimbursement invoices |
| | *Cost of adult vaccine in Zengcheng* | 22.9 | 13 | Health clinic reimbursement invoices |
| | *Cost of child vaccine in Zengcheng* | 8.38 | 18 | Health clinic reimbursement invoices |
| | *Cost of adult vaccine in Tianhe#* | 12.2 | 41 | Health clinic reimbursement invoices |
| | | Cost per item / batch (USD) | | |
| | *Surgical gloves* | 0.075 | 228 | Personal communication with research staff |
| | *Information pamphlet for free vaccination arm* | 20.4 | 1 | Project research staff invoices |

**PIF**—pay-it-forward, **USD**–United States dollars, **Hr**—Hours

* Costs were originally reported in RMB and converted to 2020 USD using the conversion rate reported by OANDA on 1 Nov 2020.

§ Hourly wages were calculated from monthly or yearly wages based on an assumption of 250 working days/year and 10 working hours/day.

† Costs marked with † were annualised over a three-year period at a discount rate of 3% as they were part of the preparatory workshop for the conception of PIF programme specifically. The training, supplies and expertise gained through these sessions are expected to be useful in future years to inform further iterations of the programme and were therefore annualised over three years.

†† It was assumed that each nurse spent 10 minutes per patient to recruit and persuade participants to be vaccinated in the pay-it-forward arm, while they spent 5 minutes per patient for the standard-of-care arm, and 5 minutes per patient in the free vaccination arm.

# Only adult influenza vaccines were used in Yangshan and Tianhe, i.e. children received the same influenza vaccine as the adults.

regarding influenza vaccination, but did not receive any community-created messages about the pay-it-forward programme. They also received their vaccination at no charge.

The questionnaire requested information about socio-demographic details of participants. The primary outcome of the parent trial was influenza vaccine coverage as assessed by administrative records. Additional outcomes included donation status (within the pay-it-forward arm) and amount donated.

*Addendum: Definitions for total economic cost and total financial cost of each intervention arm*

**Total economic cost** of an intervention: <u>*reflecting total value of resources used*</u>

= Start-up costs + Fixed costs + Recurrent costs

**Total financial cost** of an intervention: <u>*reflecting net provider cost*</u>

= Start-up costs + Fixed costs + Recurrent costs – (Payment / Donation contributions)

## Key modelling assumptions

For this study, a decision tree model was built in Excel to estimate the incremental cost of implementing a seasonal influenza vaccination programme under a pay-it-forward strategy compared with a standard-of-care strategy and a free vaccination strategy, based on costing the resources used in the quasi-experimental study, from a healthcare provider perspective.

Vaccine protection against influenza was expected to last around twelve months as influenza strains vary by season and by year [18–20]. Therefore, the benefit of the intervention did not carry over to subsequent years. A time horizon of one year was hence used to compare the costs between the three study arms.

In this analysis, vaccination uptake, i.e. the number of people vaccinated in each intervention arm, was chosen as the effectiveness measure of the pay-it-forward intervention. Incremental costs and incremental cost-effectiveness ratios (ICERs) were estimated by comparing pay-it-forward against the next most effective strategy: standard-of-care or free vaccination. Both financial and economic costs were estimated.

A micro-costing approach was used in this economic evaluation. Cost inputs for the model were obtained from information publicly available on local government websites, self-reported costs and research invoices. The unit costs of the sundry items used in each intervention arm were estimated using primary sources such as invoices and self-reported costs from the research staff who carried out the data collection, while secondary sources such as official government reports were used to estimate the wages of healthcare workers in China in the calculation of staff time cost. A currency conversion rate of 6.68 RMB to 1 USD was used, which was the conversion rate reported on OANDA on 1st November 2020, the mid-point of the data collection period [21]. Details of the unit costs of the resources involved in the model and the frequency of their use are shown in Table 1. The implementation cost of each study arm was separated into start-up, fixed and recurrent costs. Start-up costs included training costs for setting up each vaccination programme. Fixed costs consisted of the baseline cost of running the clinic, and the recurrent costs mainly included the recruitment time that nurses spent during initial consultation in order to recruit patients into the pay-it-forward programme and donate. A more detailed explanation of the actual workflow of the pay-it-forward programme and their contributions to the total cost can be found in S2 Appendix. The cost summation for financial and economic costs of each study arm is also listed in the addendum of Table 1.

## Uncertainty analyses

Deterministic sensitivity analysis (DSA) was undertaken to evaluate the impact of uncertainty around the input parameters as well as the robustness of the model. A one-way DSA was conducted to evaluate the impact of the variability in each individual parameter on the ICER.

Parameters were varied to their upper and lower bounds of the 95% CIs one at a time with all other parameters remaining at their baseline values. Cost data which did not have 95% confidence intervals (CIs) nor standard errors (SEs) were varied by +/- 30% of their reported cost given by the participating health clinics. For other variables such as vaccine uptake, their 95% CIs were taken to be their upper and lower bounds. Multi-way DSA was performed to estimate the ICERs in the worst- and best-case scenarios.

A probabilistic sensitivity analysis (PSA) was undertaken to evaluate the impact of uncertainty around the parameters and robustness of the model. A Monte Carlo simulation with 10,000 iterations was used to assess the joint impact of the uncertainty on the model output. In each iteration, each input parameter was simultaneously varied along a pre-defined probability distribution. The input parameters of the decision tree model varied in the probabilistic sensitivity analysis are shown in Table 2.

Ethical approval for this study was obtained the Zhuhai Center for Disease Control (approval number 2020011). Online consent was obtained from guardians of children and

**Table 2. Input parameters of the model varied in probabilistic sensitivity analysis.**

| Parameter | Value* (USD) | Input value (RMB) | Standard error | Distribution |
|---|---|---|---|---|
| • Monthly wage of research fellow | 2650 | 17700 | 2709 | Gamma (α = 42.7, β = 415) |
| • Monthly wage of research assistant | 1557 | 10400 | 1592 | Gamma (α = 42.7, β = 244) |
| • Monthly wage of nurse | 1003 | 6700 | 1026 | Gamma (α = 42.7, β = 157) |
| • Monthly wage of clinic coordinator | 1003 | 6700 | 1026 | Gamma (α = 42.7, β = 157) |
| • Yearly wage of vaccinators | 27695 | 185000 | 28316 | Gamma (α = 42.7, β = 4334) |
| • Cost of adult vaccine in Yangshan | 22.9 | 153 | 23.4 | Gamma (α = 42.7, β = 3.58) |
| • Cost of adult vaccine in Zengcheng | 22.9 | 153 | 23.4 | Gamma (α = 42.7, β = 3.58) |
| • Cost of child vaccine in Zengcheng | 8.38 | 56 | 8.57 | Gamma (α = 42.7, β = 1.31) |
| • Cost of adult vaccines in Tianhe | 12.2 | 81.6 | 12.5 | Gamma (α = 42.7, β = 1.91) |
| • Cost of pay-it-forward pamphlets | 29.0 | 194 | 29.7 | Gamma (α = 42.7, β = 4.54) |
| • Cost of ballpoint pen (per pen) | 0.749 | 5 | 0.765 | Gamma (α = 42.7, β = 0.117) |
| • Cost of medical glove (per piece) | 0.075 | 0.5 | 0.0765 | Gamma (α = 42.7, β = 0.012) |
| • Cost of pay-it-forward postcards | 7.41 | 49.5 | 7.58 | Gamma (α = 42.7, β = 1.16) |
| • Amount of pay-it-forward donations | 585 | 3908 | 598 | Gamma (α = 42.7, β = 91.6) |
| • Cost of free vaccination pamphlets | 20.4 | 136.5 | 20.9 | Gamma (α = 42.7, β = 3.20) |
| Parameter | Proportion | | Standard error | Distribution |
| • Uptake rate of adult vaccines in Yangshan (Pay-it-forward) | 0.27 | | 0.0364 | Beta (α = 40.7, β = 108) |
| • Uptake rate of adult vaccines in Zengcheng (Pay-it-forward) | 0.11 | | 0.0259 | Beta (α = 16.9, β = 132) |
| • Uptake rate of child vaccines in Zengcheng (Pay-it-forward) | 0.17 | | 0.0304 | Beta (α = 24.8, β = 124) |
| • Uptake rate of adult vaccines in Tianhe (Pay-it-forward) | 0.19 | | 0.0318 | Beta (α = 27.8, β = 121) |
| • Uptake rate of adult vaccines in Yangshan (Free vaccination) | 0.28 | | 0.0367 | Beta (α = 41.7, β = 107) |
| • Uptake rate of adult vaccines in Zengcheng (Free vaccination) | 0.087 | | 0.0230 | Beta (α = 12.9, β = 136) |
| • Uptake rate of child vaccines in Zengcheng (Free vaccination) | 0.12 | | 0.0265 | Beta (α = 17.9, β = 131) |
| • Uptake rate of adult vaccines in Tianhe (Free vaccination) | 0.27 | | 0.0364 | Beta (α = 40.7, β = 108) |
| • Uptake rate of adult vaccines in Yangshan (Standard-of-care) | 0.1 | | 0.0245 | Beta (α = 14.9, β = 134) |
| • Uptake rate of adult vaccines in Zengcheng (Standard-of-care) | 0.033 | | 0.0147 | Beta (α = 4.94, β = 143) |
| • Uptake rate of child vaccines in Zengcheng (Standard-of-care) | 0.093 | | 0.0238 | Beta (α = 13.9, β = 135) |
| • Uptake rate of adult vaccines in Tianhe (Standard-of-care) | 0.14 | | 0.0283 | Beta (α = 20.7, β = 128) |

**RMB**–Renminbi, **USD**–United States dollars

* Costs were originally reported in RMB and converted to 2020 USD using the conversion rate reported by OANDA on 1 Nov 2020 for clarity of understanding in this table. Subsequent standard errors and distributions used in the model were obtained from the input values reported in RMB.

older adults. The trial was registered in early September 2020 in Chinese Clinical Trial Registry with the number of ChiCTR2000040048. All information was entered and analysed using Microsoft Excel and STATA SE Version 16.1 (College Station, Texas).

## Results

Table 3 shows the socio-demographic characteristics of the study population. The study population was predominantly female, with a median age of 60 years old. Most participants lived with children at home, were married, and were educated up to high school level. Around 70% of the study participants had a monthly income level below 5000 RMB ($750) across the three

**Table 3. Population characteristics of study participants in the parent study (N = 450).**

| Characteristic | Intervention arm | | | p-value* |
| --- | --- | --- | --- | --- |
| | Pay-it-forward (n = 150) | Standard-of-care (n = 150) | Free vaccination (n = 150) | |
| | Proportion (%), (n) | | | |
| **Sex** | | | | |
| • Male | 28.67 (43) | 24.67 (37) | 26.67 (40) | 0.736 |
| • Female | 71.33 (107) | 75.33 (113) | 73.33 (110) | |
| **Age group** | | | | |
| • 0–30 | 10.67 (16) | 18.67 (28) | 11.33 (17) | 0.117 |
| • 31–60 | 46.00 (69) | 34.67 (52) | 39.33 (59) | |
| • >61 | 43.44 (65) | 46.67 (70) | 49.33 (74) | |
| **Living with elderly at home** | | | | |
| • No | 35.33 (53) | 58.00 (87) | 57.33 (86) | **<0.001** |
| • Yes | 64.67 (97) | 42.00 (63) | 42.67 (64) | |
| **Living with children at home** | | | | |
| • No | 5.33 (8) | 9.33 (14) | 6.00 (9) | 0.342 |
| • Yes | 94.67 (142) | 90.67 (136) | 94.00 (141) | |
| **Marital status** | | | | |
| • Living alone | 16.0 (24) | 16.0 (24) | 8.67 (13) | 0.101 |
| • Engaged or married | 84.0 (126) | 84.0 (126) | 91.33 (137) | |
| **Income group** | | | | |
| • 0–1000 RMB/month | 34.0 (51) | 38.0 (57) | 32.0 (48) | 0.905 |
| • 1000–5000 RMB/month | 40.0 (60) | 34.0 (51) | 38.7 (58) | |
| • 5000–10000 RMB/month | 17.3 (26) | 18.0 (27) | 18.0 (27) | |
| • >10000 RMB/month | 8.67 (13) | 10.0 (15) | 11.3 (17) | |
| **High income status** | | | | |
| • <5000 RMB/month | 74.0 (111) | 72.0 (108) | 70.7 (106) | 0.810 |
| • >5000 RMB/month | 26.0 (39) | 28.0 (42) | 29.3 (44) | |
| **Employment status** | | | | |
| • Unemployed | 52.7 (79) | 48.7 (73) | 54.7 (82) | 0.571 |
| • Employed | 47.3 (71) | 51.3 (77) | 45.3 (68) | |
| **Education group** | | | | |
| • High school or below | 65.3 (98) | 78.0 (117) | 68.7 (103) | 0.044 |
| • Bachelor's and above | 34.7 (52) | 22.0 (33) | 31.3 (47) | |
| **Vaccination status** | | | | |
| • Refused vaccine | 26.0 (39) | 63.3 (95) | 24.0 (36) | **<0.001** |
| • Received vaccine | 74.0 (111) | 36.7 (55) | 76.0 (114) | |

* Obtained from the Pearson chi-square test of independence

**Table 4. Outcome analyses for costs of each intervention arm.**

| Intervention arm | Total costs: i.e. resource use (USD) | User contributions | Net provider costs | Number of people vaccinated | Average cost per person vaccinated | Incremental number of people vaccinated | ICER (USD per person vaccinated) using financial cost | ICER (USD per person vaccinated) using economic cost |
|---|---|---|---|---|---|---|---|---|
| Standard-of-care | 3557 | 832 | 2725 | 55 | 49.55 | - | - | - |
| Pay-it-forward | 5062 | 585 | 4477 | 111 | 40.33 | 56 | 31.29 | 26.88 |
| Free vaccination | 4665 | 0 | 4665 | 114 | 40.92 | 3 | 62.67 | -132.33 |

**ICER** = incremental cost-effectiveness ratio, **USD** = United States dollars

study arms. There were differences in the distribution of age, income level, and education level between the three study sites, as well as in the factors of living with children and living with elderly at home.

We present the results of the PSA (Table 2) in a cost-effectiveness plane and cost-effectiveness acceptability curves (CEACs). These curves plot the proportion of the incremental cost-effect pairs that are cost-effective for a range of values of the willingness-to-pay threshold, which in this study ranged from $0-$110 per additional person vaccinated, comparable to the range of ICERs of $0.66-$161.95 (in 2017 USD) per child vaccinated reported in a recent systematic review [22].

Table 1 shows the itemised costing breakdown of items for each study arm and details how the financial and economic costs were defined. The financial and economic costs of each intervention arm, their incremental costs, incremental number of people vaccinated, and their ICERs are reported in Table 4. Comparing pay-it-forward to standard of care using financial costs, the financial cost required for vaccinating an additional person was $31.29. When considering the ICER from an economic cost perspective, which now includes donations or volunteer time, the incremental economic cost of implementing influenza vaccination for standard of care increased to $3557. The economic cost of the same intervention under a pay-it-forward strategy increased to $5062, and the economic cost under a free vaccination strategy increased to $4665. Pay-it-forward was dominated by free vaccination as it became both more costly and marginally less effective in comparison (111 vs 114 vaccinations). Cost offsets further lower the cost of the pay-it-forward model. However, when comparing pay-it-forward directly with standard-of-care using economic costs, the ICER was $26.88/person vaccinated, which was an improvement from the ICER obtained using financial costs ($26.88 vs $31.29/person vaccinated).

When one-way DSA was performed, the most important factor affecting the ICER obtained with financial costs was the difference in vaccine uptake between the pay-it-forward and standard-of-care arms, which shifted the upper limit of the ICER to $77.19/person vaccinated (Fig 3A). At the lower limit, i.e. when the vaccine uptake of pay-it-forward greatly outstripped that of standard-of-care, this resulted in significant improvement of the ICER to $19.62/person vaccinated. The one-way DSA using economic costs showed similar results (Fig 3B). This shows that the current intervention is only effective if there is a substantial difference in vaccine uptake of pay-it-forward and standard-of-care arms and has the potential to be even more cost-effective in future iterations if a higher vaccine uptake rate could be achieved in pay-it-forward.

When multi-way DSA was performed, the worst-case scenario generated an ICER of $45.68/person vaccinated when comparing pay-it-forward against standard-of-care using

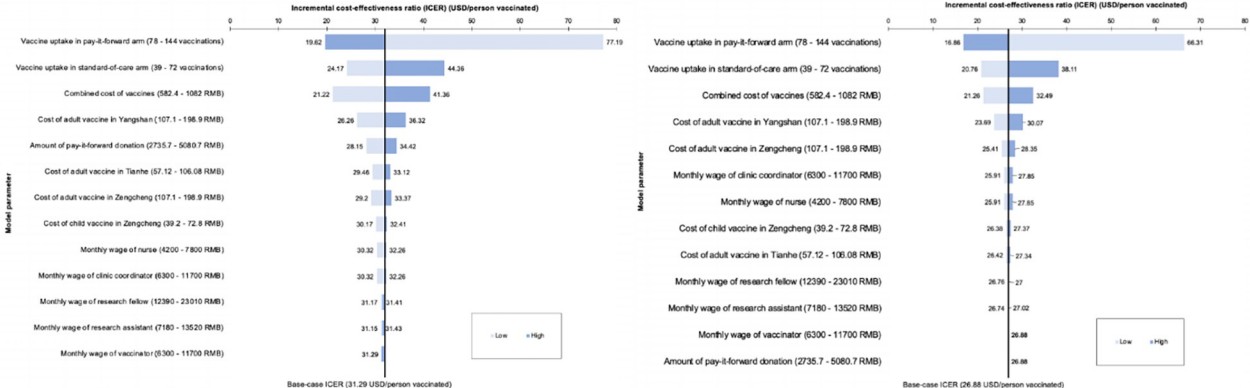

**Fig 3. a:** Tornado diagram showing the effect of varying different input parameters on the ICER comparing pay-it-forward against standard-of-care calculated using financial costs, **b:** Tornado diagram showing the effect of varying different input parameters on the ICER comparing pay-it-forward against standard-of-care calculated using economic costs.

financial costs. The best-case scenario generated an ICER of $21.26/person vaccinated when comparing pay-it-forward against standard-of-care, considering financial costs only. The ICER obtained when moving from pay-it-forward to the next best intervention (free vaccination) was $63.00/person vaccinated. When considering economic costs, pay-it-forward was still dominated by free vaccination in both best- and worst-case scenarios.

When PSA was performed, all simulations predicted that pay-it-forward would result in a higher number of vaccinations (Fig 4) but would cost more than the current standard-of-care practice when considering only financial costs. Fig 5 shows the same comparison but considering only economic costs, which also predicted similarly that pay-it-forward would be more

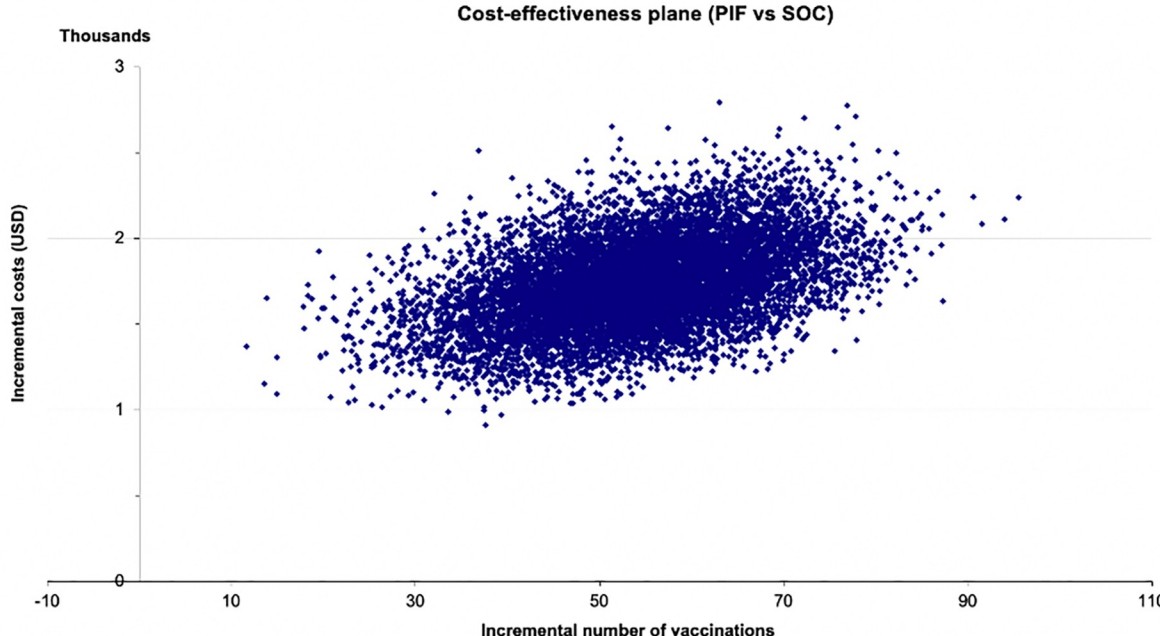

**Fig 4. Cost-effectiveness plane showing the incremental costs and incremental number of vaccinations of comparing pay-it-forward with standard-of-care using financial costs.**

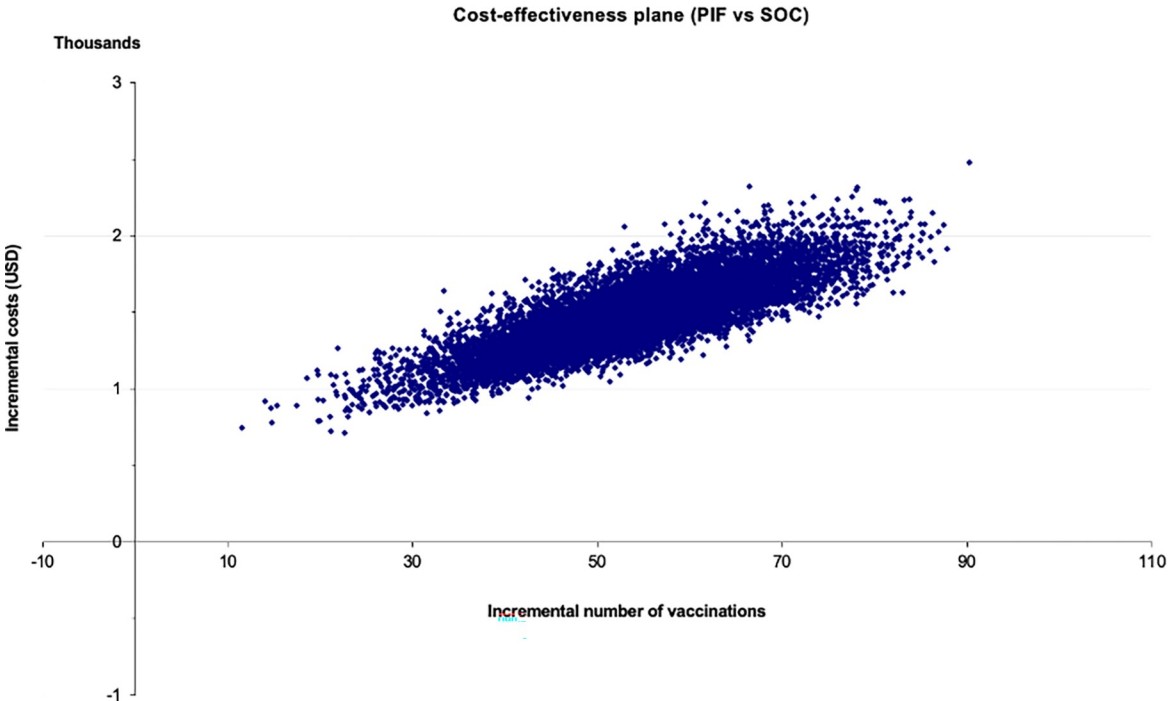

**Fig 5. Cost-effectiveness plane showing the incremental costs and incremental number of vaccinations comparing pay-it-forward with standard-of-care using economic costs.**

costly but more effective than standard-of-care. The average ICER comparing pay-it-forward with standard-of-care using financial costs was $33.2/person vaccinated (95%CI $24.0–$45.9). The average ICER comparing pay-it-forward with standard-of-care using economic costs was $28.0/person vaccinated (95%CI $23.1–$34.4). These ICERs were similar to the original base case estimates ($31.3/person vaccinated and $26.9/person vaccinated).

Figs 6 and 7 show the probability of each intervention being cost-effective at different willingness-to-pay thresholds using economic and financial costs respectively.

When comparing pay-it-forward against standard-of-care using economic costs, starting at a willingness-to-pay (WTP) threshold of $20/person vaccinated, pay-it-forward was cost-effective only in 19.5% of simulations, rising sharply to 75.9% of simulations at a WTP threshold of $30/person vaccinated, then to 99.1% of simulations when the WTP threshold further increased to $40/person vaccinated. Similarly, when comparing these two interventions using financial costs, pay-it-forward was cost-effective in 86.3% of simulations at a WTP threshold of $40/per vaccinated.

When comparing pay-it-forward against free vaccination using financial costs, since the degree of uncertainty in the incremental costs and incremental effects between these two groups was large, even at a WTP threshold of $0/person vaccinated, there was some probability (18.1%) that free vaccination could be cost-effective compared to pay-it-forward. The probability of being cost-effective increased gradually with increase in the WTP threshold, but remained below 90% even at a WTP threshold of $110/person vaccinated.

## Discussion

This study provides a partial economic evaluation of an innovative pay-it-forward approach to improve influenza vaccine uptake. Our findings suggest that the pay-it-forward approach was

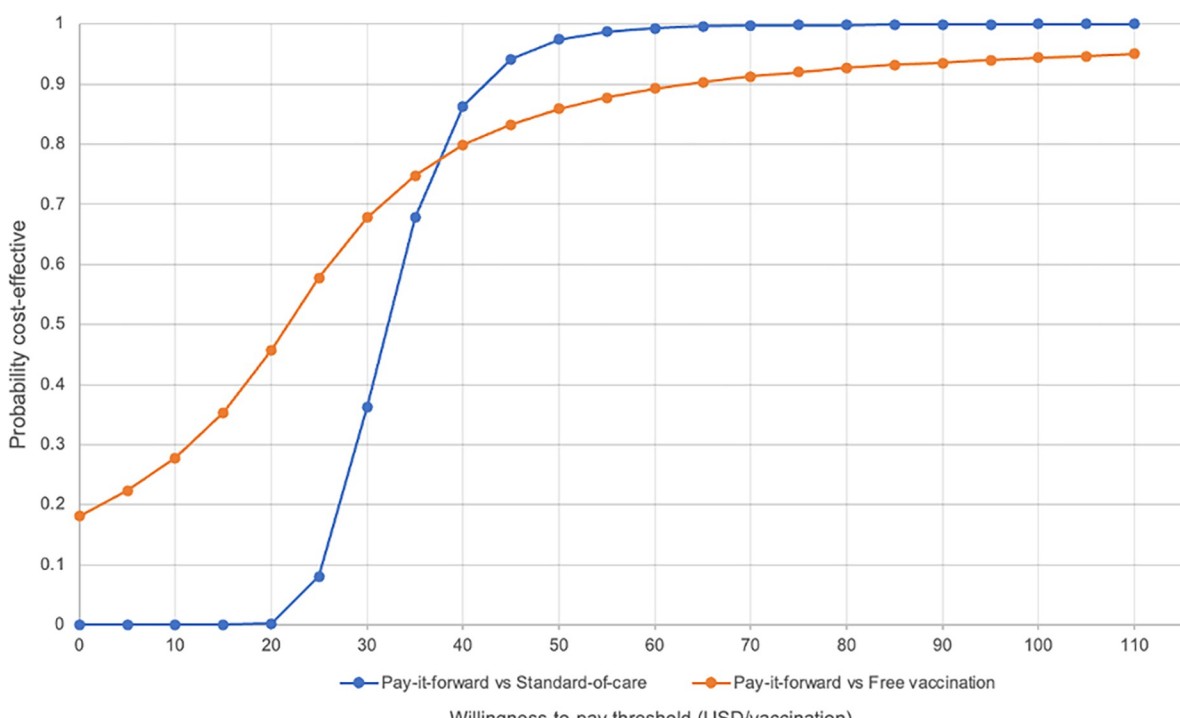

**Fig 6. Cost-effectiveness acceptability curve showing the probability of pay-it-forward and free vaccination being cost-effective under different willingness-to-pay thresholds when considering financial costs.**

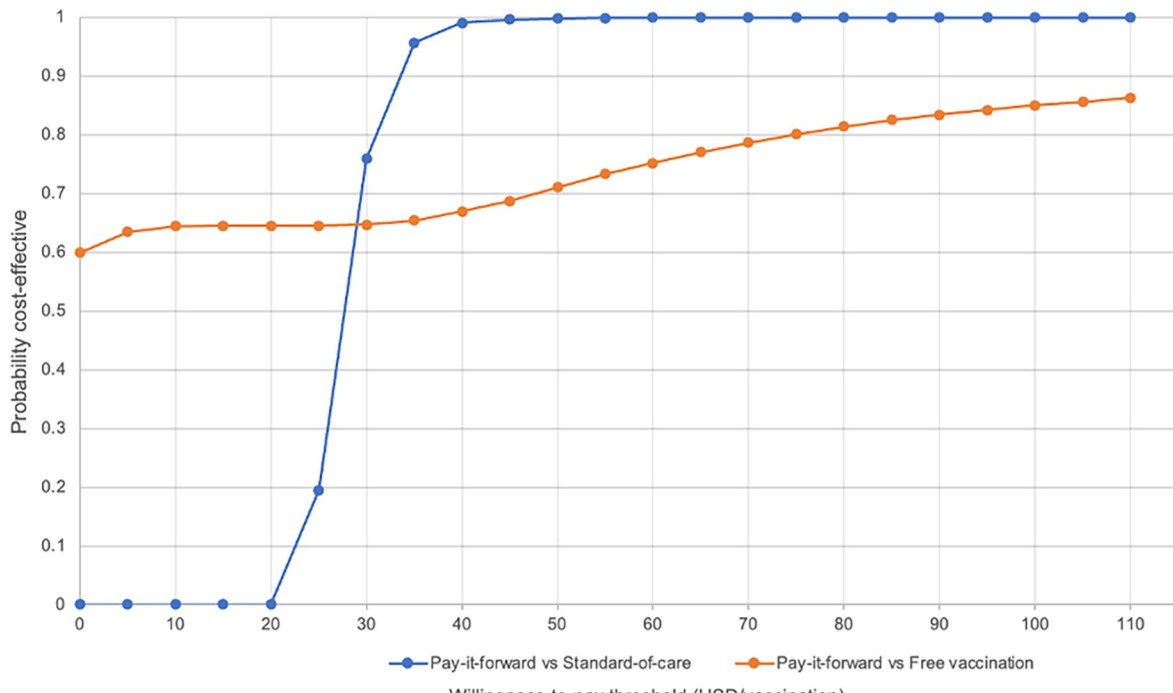

**Fig 7. Cost-effectiveness acceptability curve showing the probability of pay-it-forward and free vaccination being cost-effective under different willingness-to-pay thresholds when considering economic costs.**

more effective and more costly than the standard-of-care vaccination approach. However, the pay-it-forward and free vaccination approaches had lower costs per person vaccinated compared to standard-of-care. This study expands the literature by examining the cost-effectiveness of a social innovation and examining the donations that are critical to the pay-it-forward approach.

Our data suggest that pay-it-forward strategies could be useful to increase vaccination uptake. Our study recruited older adults and caregivers who may not have as strong of a sense of community identity compared to sexual minorities who participated in past iterations of pay-it-forward interventions [23]. Pay-it-forward may increase vaccine uptake because of the free testing component, the community engagement, or both of these elements.

Our data suggest that the incremental financial cost per person vaccinated was $31.29. This vaccine cost is comparable to costs of implementation strategies to improve infant immunisation coverage in LMICs ($41.89) [24]. The financial cost per person vaccinated in the standard of care arm was $49.55, while the value for the pay-it-forward arm was $40.33. The financial cost per person for the free vaccination arm was $40.92. Based on cost per person alone, the pay it forward arm shows to be the least expensive option per additional person vaccinated for vaccination implementation. It is important to note this was only a preliminary pilot and there was no experimental implementation to enhance donations.

The inclusion of encouragement to community members to write a note to future vaccine recipients mobilizes community engagement, which can improve vaccine confidence. Receiving positive social feedback by reading supportive messages from fellow community members can add to the vaccine confidence effect [25].

Previous pay-it-forward studies generally reported high donation rates but low average donation amounts [26]. Thus, there is a need to enhance donation amounts in future pay-it-forward programmes to improve financial sustainability and for consideration of scaling-up. Since data regarding factors influencing donations in pay-it-forward is sparse, this study would add to previous studies and help identify the drivers behind donation amount.

Our study has several limitations. First, this is only based on a single cross-sectional quasi-experimental study. However, the study included people from three different socio-economic settings in Guangdong, China. The study population was diverse in income level and education status. The study also targeted both older adults and children, which allowed us to explore its effectiveness in improving vaccine uptake among different age groups. Second, the study was implemented during the COVID-19 pandemic. However, the relative effectiveness of the intervention (number vaccinated per eligible participant) was relatively consistent during the lockdown period and after lifting lockdown. This analysis took a healthcare service provider perspective, which was narrower compared to a societal perspective. The study did not capture non-health and non-material benefits received by the local community through pay-it-forward, a significant part of the intervention. Our study was not designed to assess the inter-site differences in cost-effectiveness. However, the total amount donated and the vaccine uptake was similar across sites. Additionally, under local social distancing measures, postcard gifting from one participant to another were not allowed in some study sites. These non-health benefits would likely increase the cost-effectiveness of the intervention. Several downstream health benefits of increased seasonal influenza vaccination like direct health effect, herd immunity and reduced burden on the healthcare system have also not been captured in this study.

Vaccination is frequently undervalued as the standard framework of economic evaluation of health technologies is often insufficient to reflect the full range of health and economic benefits conferred by vaccines [27]. When implemented on a large scale, the pay-it-forward program can utilize some financial cooperation from community actors to keep cost low. Further analysis is needed understand how the approach could be used to support vulnerable groups

and leverage prosocial momentum. The scope of our analysis on the pay-it-forward model could be broadened to allow for a more comprehensive view of its benefits. Some potential aspects of analysis may include direct health effect or more downstream macroeconomic consequences. In order to effectively implement pay- it-forward operates, more must be known about the mechanisms of why pay-it-forward works through implementation and mixed methods research. Policy innovations like pay-it-forward are needed to decrease financial barriers of vaccine uptake. This could be attractive to policymakers as a way to support vaccine campaigns with fewer public resources.

## Supporting information

**S1 Appendix. English version of questionnaire administered to study participants.**
(DOCX)

**S2 Appendix. Flow and organisation of pay-it-forward programme implementation in this study.**
(DOCX)

**S1 Data. PIF influenza participant questionnaire responses and donation amounts dataset.**
(XLSX)

**S2 Data. PIF influenza costing analysis and probabilistic sensitivity analysis.**
(XLS)

## Author Contributions

**Conceptualization:** Fanny Fong-Yi Tang, Dan Wu, Jason J. Ong.

**Data curation:** Fanny Fong-Yi Tang, Jason J. Ong.

**Formal analysis:** Fanny Fong-Yi Tang.

**Funding acquisition:** Dan Wu, Jason J. Ong.

**Investigation:** Fanny Fong-Yi Tang, Dan Wu, Jason J. Ong, Joseph D. Tucker.

**Methodology:** Fanny Fong-Yi Tang, Joseph D. Tucker.

**Project administration:** Dan Wu.

**Resources:** Dan Wu.

**Supervision:** Dan Wu, Jason J. Ong, Joseph D. Tucker.

**Validation:** Dan Wu.

**Visualization:** Fanny Fong-Yi Tang, Fern Terris-Prestholt.

**Writing – original draft:** Fanny Fong-Yi Tang.

**Writing – review & editing:** Priya Kosana, Mark Jit, Fern Terris-Prestholt, Joseph D. Tucker.

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
