## [Decision Letter · Decision Letter 0]

31 May 2023

PGPH-D-23-00017

Pay-it-forward influenza vaccination among older adults and children: A cost-effectiveness analysis in China

Dear Dr. Kosana,

Thank you for submitting your manuscript to PLOS Global Public Health. After careful consideration, we feel that it has merit but does not fully meet PLOS Global Public Health’s publication criteria as it currently stands. Therefore, we invite you to submit a revised version of the manuscript that addresses the points raised during the review process.

The reviewers brought up minor issues. Please respond to each. If you do not think an edit is warranted, please explain why.

We look forward to receiving your revised manuscript.

Kind regards,

Abram L. Wagner, PhD, MPH

Academic Editor

Journal Requirements:

3. Please provide separate figure files in .tif or .eps format only and remove any figures embedded in your manuscript file. Please also ensure all files are under our size limit of 10MB.

Additional Editor Comments (if provided):

Reviewers' comments:

Reviewer's Responses to Questions

**Comments to the Author**

1. Does this manuscript meet PLOS Global Public Health’s publication criteria? Is the manuscript technically sound, and do the data support the conclusions? The manuscript must describe methodologically and ethically rigorous research with conclusions that are appropriately drawn based on the data presented.

Reviewer #1: Yes

Reviewer #2: Yes

2. Has the statistical analysis been performed appropriately and rigorously?

Reviewer #1: Yes

Reviewer #2: Yes

3. Have the authors made all data underlying the findings in their manuscript fully available (please refer to the Data Availability Statement at the start of the manuscript PDF file)?

Reviewer #1: Yes

Reviewer #2: No

4. Is the manuscript presented in an intelligible fashion and written in standard English?

Reviewer #1: Yes

Reviewer #2: Yes

5. Review Comments to the Author

Reviewer #1: a) Minor comment: Page 6: Last paragraph under ‘’introduction’’: Correct the grammar from ‘’ We conducted a three-arm quasi-experimental study is to assess the effectiveness of….’’ To ‘’ We conducted a three-arm quasi-experimental study to assess the effectiveness of…..’’

b) What were the eligibility/inclusion criteria for each experimental arm?

c) At which minimum vaccine uptake threshold will it not be cost effective in the pay-it-forward arm and vice versa when comparing with the standard care?

d) What is the health policy implication of your major finding that pay-it-forward approach was more effective and more costly than the standard-of-care vaccination approach? What opportunities exist to keep the cost low at the community level?

e) Were there inter-site differences in cost-effectiveness across the three study sites and if any, were the differences across the arms per site significant?

Reviewer #2: The study evaluates the cost-effectiveness of a "Pay-It-Forward" vaccination program. This program involves participants receiving a free influenza vaccination and being asked if they would like to donate or create a message to support subsequent vaccinations. The authors provide a detailed account of how the Pay-It-Forward program was implemented, including a preparatory workshop, the design of postcards, the hiring of doctors as vaccinators and quality checkers, and the recruitment time of nurses. The program was found to be more effective, it spurred 96.4% of individuals to voluntarily donate to support influenza vaccination for high-risk groups in China.

This study provides valuable insights into alternative methods of promoting and funding vaccination programs.

6. PLOS authors have the option to publish the peer review history of their article (what does this mean?). If published, this will include your full peer review and any attached files.

**Do you want your identity to be public for this peer review?** For information about this choice, including consent withdrawal, please see our Privacy Policy.

Reviewer #1: No

Reviewer #2: No

---

## [Decision Letter · Decision Letter 1]

12 Jul 2023

Pay-it-forward influenza vaccination among older adults and children: A cost-effectiveness analysis in China

PGPH-D-23-00017R1

Dear Ms Tang,

We are pleased to inform you that your manuscript 'Pay-it-forward influenza vaccination among older adults and children: A cost-effectiveness analysis in China' has been provisionally accepted for publication in PLOS Global Public Health.

Best regards,

Abram L. Wagner, PhD, MPH

Academic Editor

Reviewer Comments (if any, and for reference):

Reviewer's Responses to Questions

**Comments to the Author**

1. If the authors have adequately addressed your comments raised in a previous round of review and you feel that this manuscript is now acceptable for publication, you may indicate that here to bypass the “Comments to the Author” section, enter your conflict of interest statement in the “Confidential to Editor” section, and submit your "Accept" recommendation.

Reviewer #1: All comments have been addressed

2. Does this manuscript meet PLOS Global Public Health’s publication criteria? Is the manuscript technically sound, and do the data support the conclusions? The manuscript must describe methodologically and ethically rigorous research with conclusions that are appropriately drawn based on the data presented.

Reviewer #1: Yes

3. Has the statistical analysis been performed appropriately and rigorously?

Reviewer #1: Yes

4. Have the authors made all data underlying the findings in their manuscript fully available (please refer to the Data Availability Statement at the start of the manuscript PDF file)?

Reviewer #1: Yes

5. Is the manuscript presented in an intelligible fashion and written in standard English?

Reviewer #1: Yes

6. Review Comments to the Author

Reviewer #1: Congratulations to the research team.

7. PLOS authors have the option to publish the peer review history of their article (what does this mean?). If published, this will include your full peer review and any attached files.

**Do you want your identity to be public for this peer review?** For information about this choice, including consent withdrawal, please see our Privacy Policy.

Reviewer #1: **Yes: **Dr. Richard AMENYAH
